# Analysis of CD1a-Positive Monocyte-Derived Cells in the Regional Lymph Nodes of Patients with Gallbladder Cancer

**DOI:** 10.3390/ijms252312763

**Published:** 2024-11-27

**Authors:** Sachiko Maeda, Keita Kai, Kanako Kawasaki, Tomokazu Tanaka, Takao Ide, Hirokazu Noshiro

**Affiliations:** 1Department of Pathology & Microbiology, Faculty of Medicine, Saga University, Saga 849-8501, Japan; 2Department of Surgery, Faculty of Medicine, Saga University, Saga 849-8501, Japan; f8642@cc.saga-u.ac.jp (T.T.); idetaka@cc.saga-u.ac.jp (T.I.); noshiro@cc.saga-u.ac.jp (H.N.); 3Department of Pathology, Saga University Hospital, Saga 849-8501, Japan

**Keywords:** dendritic cell, CD1a, monocyte, lymph node, gallbladder cancer

## Abstract

Dendritic cells (DCs) are known to be major antigen-presenting cells, and lymph nodes (LNs) play an important role in DC-mediated immune response. CD1a is known as a marker of monocyte-derived DCs. The present study focused on the infiltration of CD1a-positive DCs (CD1a-DCs) into regional LNs in 70 cases of gallbladder cancer (GBC). After univariate analyses, the results showed that LN infiltration by CD1a-DCs was associated with unfavorable clinical outcomes in patients with GBC, with all cases categorized in the CD1a-DCs high group had nodal metastasis. LN infiltration by CD1a-DCs was not an independent prognostic factor identified by multivariate analyses. After subgroup analyses of cases with LN metastasis (*n* = 32), no significant impacts of CD1a-DCs infiltration into metastatic LNs were observed. In contrast, CD1a-DCs infiltration into primary tumors had a significant impact on surgical outcomes. The results of strong confounding between CD1a-DCs and LN metastasis support the theory that CD1a-DCs are developed from monocytes at tumor sites. As the results of previous research focused on CD1a-DCs infiltration into regional LNs of other organs varied, the role and significance of CD1a-DCs infiltration in regional LNs may be different according to the tumor histology or its primary site. Thus, further studies are needed to clarify the role and significance of CD1a-DCs infiltration into regional LNs of solid cancers.

## 1. Introduction

Gallbladder cancer (GBC) is an aggressive malignancy with a poor prognosis. The 5-year survival rate of GBC is 39.8% and the 5-year survival rates per tumor stage are 70.9% in Stage II, 30.4% in Stage IIIA, 29.7% in Stage IIIB, 7.3% in Stage IVA and 4.1% in Stage IVB [1]. Patients with early-stage GBC unfortunately rarely exhibit symptoms, and diagnosis by imaging modality is often difficult to make [2], with many GBC cases being detected later with advanced cancer. Only about 19.8% of GBC cases are diagnosed in Stage I [1]. However, there is no established effective treatment for unresectable and recurrent GBC at the present time.

Dendritic cells (DCs) are a highly heterogeneous cell population comprised of several subsets with distinct origins, locations, markers, migratory and functional properties [3]. Conventional DCs (cDCs) are specialized antigen-processing and presenting cells, equipped with high phagocytic activity as immature cells and high cytokine-producing capacity as mature cells [4]. DCs usually exist in two different functional states, mature and immature. These are distinguished by many features, but the ability to activate antigen-specific naïve T cells in secondary lymphoid organs is the hallmark of mature DCs. Immature DCs are poor inducers of naïve lymphocyte effector responses and do not release immunostimulatory cytokines. Indeed, immature DCs act not only as sentinels against invading pathogens, but also as tissue scavengers, capturing apoptotic and necrotic cells [5].

DCs, as key activators of the adaptive immune response, would be expected to have a central role in inducing antitumor immune responses, and the many functional actions of these cells shown in cancer patients emphasize the relevant roles they may, indeed, play in anti-tumor immune responses [6].

The immune microenvironment in lymph nodes (LNs) is orchestrated by immune cells including macrophages, DCs, T cells, B cells and non-immune cells, such as fibroblastic reticular cells, blood endothelial cells and lymphatic endothelial cells. One of the key regulators of the host immune system that attacks cancer cells are DCs, which are highly specialized antigen-presenting cells that play crucial roles in the initiation of cellular immunity [7].

Among the various functional attributes of DCs, the endogenous migration of DCs from antigen encounter or tumor sites, or in the case of ex vivo-generated DC vaccines, their migration from the injection site to draining LNs is critical, as it permits DCs to carry out their most important role of interacting with and activating adaptive immune cells [7].

CD1 molecules are structurally related to MHC I and, like them, present antigen to αβ and γδ T cells. The 5 CD1 isoforms in humans fall into 3 groups on the basis of sequence homology and immune functions. Group 1 contains CD1a, CD1b and CD1c, which present antigens to clonally diverse T cells. CD1a presents a broad array of lipid antigens, making it a highly versatile mediator of many different immune responses [7] and as markers for immature dendritic cells such as Langerhans cells present in human skin.

Human DCs are divided into plasmacytoid DCs (pDCs) expressing surface markers CD123, blood dendritic cell antigen (BDCA)-2 (CD303), BDCA-4 (CD304), conventional DCs (cDCs), and monocyte-derived DCs (moDCs) [8,9]. As moDCs have characteristics of both DCs and macrophages, moDCs and monocyte-derived macrophages (moMacs) can be difficult to distinguish [10]. In the discussion regarding distinguishing moDCs and moMacs, CD1a is considered as a marker of moDCs [9,11]. Morphological analysis has proved to be a reliable method for distinguishing moDCs from moMacs. moMacs have an extensive cytoplasm containing numerous phagocytic vacuoles, whereas moDCs are smaller and possess dendrites on their surface, similar to the canonical description of DCs [11,12,13,14,15].

Previous reports have indicated that tumor infiltration by CD1a positive DCs (CD1a-DCs) was associated with favorable clinical outcomes in GBC [16]. Several studies have suggested that the infiltration of DCs into LNs was associated with favorable clinical outcomes in carcinomas of the breast [17], oral squamous cell carcinoma [18] and laryngeal cancer [19]. To the best of our knowledge, there have been no reports published that focused on DCs infiltration into regional LNs of GBC. Thus, the aim of the present research was to clarify the status as well as the clinical impact of CD1a-DC infiltration into regional LNs in patients with GBC. We also investigated S-100 positive DCs and CD209 (DC-SIGN) positive DCs infiltration into the regional LNs of GBC.

## 2. Results

### 2.1. Patients and Assessment of DCs in Regional LNs of Patients with GBC

The clinicopathological features of patients with GBC (*n* = 70) and the results of the assessment of DCs in regional LNs and primary tumor (CD1a-DCs only) are summarized in Table 1. The mean patient age at the time of surgery was 71.1 ± 8.8 years; 29 cases (41.4%) were male, and 41 cases (58.6%) were female, and 32 cases had LN metastasis. During the assessment of CD1a-DCs into regional LNs, 14 cases (20.0%) were classified as the high CD1a-DCs group, and the remaining 56 cases (80.0%) were classified as the low CD1a-DCs group. During the assessment of S100-DCs in regional LNs, 45 patients (64.3%) were classified into the high S100-DCs group, and the remaining 25 patients (35.7%) were classified into the low S100-DCs group. Representative histological illustrations of CD1a-DCs and S100-DCs in regional LNs are shown in Figure 1. Regarding CD209 (DC-SIGN), many CD209-positive cells were found at the sinuses of LNs, both metastatic LNs and non-metastatic LNs (Figure 2) and therefore comparative analyses of CD209-positive cells were difficult. Regarding the CD1a-DCs in primary tumors, 35 cases (50.0%) were classified as the high CD1a-DCs group, and the remaining cases were classified as the low CD1a-DCs group. Adjuvant therapy was performed in 37 cases (52.9%).

### 2.2. Association Between Clinicopathological Features and the Status of DCs in the Regional LNs of Patients with GBC

The association between the clinicopathological features and status of CD1a-DCs in regional LNs are summarized in Table 2. The advanced T-stage group (T3/T4) was significantly correlated with the high CD1a-DCs group (*p* = 0.017). Significant association was also observed between the high CD1a-DCs group and LN metastasis (*p* < 0.000). Of note, all high CD1a-DCs cases had LN metastasis and CD1a-DCs were observed at the nearby metastatic tumor. No high-density CD1a-DCs infiltration was observed in LNs without lymph node metastasis. We assessed the infiltration of CD1a-DCs into primary tumor tissue as documented in our previous study [10] but no significant association was found between CD1a-DCs infiltration into the primary tumor and LNs.

The association between the clinicopathological features and status of the S100-DCs in regional LNs are summarized in Table 3. In contrast to CD1a-DCs, there was no significant relationship between the clinicopathological features (age, sex, T-stage, LN metastasis, distant metastasis, and status of adjuvant therapy) and the degree of S100-DCs infiltration into regional LNs.

### 2.3. Kaplan-Meier Survival Curves According to the Infiltration of CD1a- and S100-DCs into Regional LNs

The survival curves for OS, DSS and RFS, according to the status of CD1a-DC and S100-DC infiltration into regional LNs, are shown in Figure 3. The high CD1a-DC in the LNs group showed a significantly worse OS (*p* = 0.028), worse DSS (*p* = 0.010) and worse RFS (*p* = 0.020) than for patients in the low CD1a-DC in the LNs group. In contrast, the status of S100-DC infiltration into LNs had no detectable significant impact on OS, DSS and RFS.

### 2.4. Univariate Analyses for DSS, OS and RFS in All Patients (n = 70)

The results of univariate analysis of potential factors associated with OS, DSS and RFS are summarized in Table 4. The factors significantly associated with OS were T stage (T1/T2 vs. T3/T4; *p* < 0.000), N stage (N0 vs. N1/N2; *p* = 0.001), M stage (M0 vs. M1; *p* = 0.003) and CD1a-DCs in LNs (high vs. low; *p* = 0.028).

The factors significantly associated with DSS were T stage (T1/T2 vs. T3/T4; *p* < 0.000), N stage (N0 vs. N1/N2; *p* = 0.000), M stage (M0 vs. M1; *p* = 0.000), CD1a status in tumor (high vs. low; *p* = 0.041) and CD1a-DCs status in LN (high vs. low; *p* = 0.010). Factors associated with RFS were T stage (T1/T2 vs. T3/T4; *p* = 0.001), N stage (N0 vs. N1/N2; *p* = 0.001), M stage (M0 vs. M1; *p* < 0.000), CD1a-DCs status in LN (high vs. low; *p* = 0.020) and adjuvant therapy (done vs. not done).

### 2.5. Multivariate Analyses of Factors Associated with OS, DSS and RFS (n = 70)

The results of multivariate analyses are listed in Table 5. The only factor significantly associated with OS and DSS was T-stage, and the only factor significantly associated with RFS was M stage. The CD1a-DC status in LNs had no significant impact on OS, DSS and RFS, due to confounding of the N stage (high-density CD1a-DCs infiltration was only observed in metastatic LNs).

### 2.6. Subgroup Analyses in Cases in the LN Metastasis Group

To confirm the impact of CD1a-DC infiltration into metastatic LNs, we performed the subgroup analyses for cases with LN metastasis (*n* = 32). The survival curves for OS, DSS and RFS according to the status of CD1a-DC infiltration into regional LNs and primary tumor are shown in Figure 4. In metastatic LNs, a significant impact of CD1a-DCs infiltration was not observed for OS, DSS or RFS. In contrast, a significant impact of CD1a-DCs infiltration into primary tumors was observed in each OS (*p* = 0.002), DSS (*p* = 0.001) and RFS (*p* = 0.001), findings consistent with our previous study results [17].

The results of univariate analysis of factors associated with OS, DSS and RFS in cases with LN metastasis are summarized in Table 6. The factors significantly associated with OS were M stage (*p* = 0.049), CD1a-DCs in tumor (*p* = 0.002) and adjuvant therapy (*p* = 0.010). The factors significantly associated with DSS were the M stage (*p* = 0.010) and CD1a-DCs in tumor (*p* = 0.001). The factors significantly associated with RFS were the M stage (*p* = 0.017) and CD1a-DCs in tumor (*p* = 0.001).

## 3. Discussion

Previous reports have indicated that infiltration of CD1a-DCs into the primary tumor is associated with favorable clinical outcomes in GBC [17]. DCs take up antigens and pathogens, generate MHC–peptide complexes, migrate from the sites of antigen acquisition to secondary lymphoid organs and, finally, they physically interact with and stimulate T lymphocytes [20]. CCR7 and its ligands are crucially important for DC migration into lymph nodes from the tumor [21,22,23]. Anti-tumoral responses depend upon a specialized subset of conventional DCs that transport tumor antigens to draining lymph nodes and cross-present antigen to activate cytotoxic T lymphocytes [24]. We initially considered that LN infiltration by CD1a-DCs would be associated with favorable clinical outcomes and that if CD1a-DCs migrated into regional LNs from the primary site, there would be CD1a-DCs in not only metastatic LNs but also in LNs without cancer metastasis.

Contrary to our expectations, the results showed that LN infiltration by CD1a-DCs was associated with unfavorable clinical outcomes in patients with GBC. The results also revealed that all cases with high CD1a-DCs numbers in LNs had LN metastasis. Therefore, the impact of LN infiltration by CD1a-DCs on unfavorable clinical outcomes is considered as confounding as LN metastasis. In fact, LN infiltration by CD1a-DCs was not an independent prognostic factor revealed by multivariate analyses of OS, DSS and RFS. In addition, subgroup analyses of cases with LN metastasis and LN infiltration by CD1a-DCs had no significant impact on OS, DSS and RFS. In contrast, CD1a-DCs infiltration into primary tumor was significantly associated with OS, DSS and RFS, findings consistent with our previous study [16].

We had initially hypothesized that the numbers of CD1a-DCs in non-metastatic sentinel LNs would not be different from that in metastatic LNs, if CD1a-DCs migrate from primary tumors to sentinel LNs. However, the number of CD1a-DCs in non-metastatic sentinel LNs was significantly fewer than that of metastatic DCs, whereas the number of S100-DCs was not different between metastatic LNs and non-metastatic LNs. These results indicate a hypothesis that a component of CD1a-DCs in metastatic LNs may differentiate from monocytes through interaction with metastatic tumors other than migration from the primary tumor to LNs. This hypothesis was supported by evidence that moDCs have a low migratory capacity and do not express CCR7, which is the chemokine receptor required for migration to LNs [11,12].

Several studies have investigated the association between CD1a-DCs infiltration into LNs and into the primary lesion of various organs, such as the breast [25,26,27,28,29,30], skin (melanoma) [31,32,33,34], larynx [19], oral cavity [18,35], stomach [36] and the uterus [37] using various experimental approaches. However, the results were at best variable, and the role and significance of CD1a-DCs infiltration into regional LNs and/or primary tumor still remains unclear. As the CD1a-DCs develop from monocytes under the conditioning of cytokines such as granulocyte-macrophage-colony-stimulating factor (GM-CSF), tumor necrosis factor (TNF)-α, interleukin (IL)-6, IL-4, and IL-1β [11,38], induction of CD1a-DCs may differ according to the tumor characteristics producing these cytokines. Our result that CD1a-DCs are significantly more induced in metastatic LNs than in non-metastatic LNs supports the theory that CD1a-DCs are developed at tumor sites.

In breast cancer, a number of studies have reported that the degree of CD1a-DCs infiltration into LNs was similar in metastatic LNs and non-metastatic LNs [25,27,28] and that CD1a-DCs infiltration into LNs had no significant impact on prognosis [25,28]. On the other hand, Rocca et al. reported that the degree of the CD1a-DCs infiltration into metastatic LNs was significantly lower than into non-metastatic LNs [28]. Yang et al. [29]. compared the baseline infiltration of CD1a-DCs between sentinel LNs and non-sentinel LNs and their results appeared to indicate that the number of CD1a-DCs was lower in sentinel LNs than in non-sentinel LNs (3.3% vs. 7.0%), although statistical significance was not reached. Zheng et al. [30] investigated the impact of CD1a-DCs infiltration into the primary lesion of triple-negative breast cancer and their results indicated a significant association between CD1a-DCs infiltration and a worse OS. The latter study also found a significant association between CD1a-DCs infiltration and PD-L1 expression in triple-negative breast cancer.

In melanoma, CD1a-DCs numbers were significantly lower in non-sentinel LNs compared to sentinel LNs [31,32] and the frequencies of CD1a-DCs subsets in sentinel LNs, assessed by flow cytometry, were related to local recurrence [33]. Barbour et al. [34] investigated CD1a-DCs infiltration into metastatic LNs of melanoma and suggested that low numbers of CD1a-DCs infiltration was correlated with reduced effector cell activation.

In oral squamous cell carcinoma, CD1a-DCs infiltration was found to be significantly higher in metastatic LNs than in non-metastatic LNs [18], whereas one study reported no significant difference in CD1a-DCs infiltration between metastatic LNs and non-metastatic LNs [35]. In laryngeal squamous cell carcinoma, no significant difference in CD1a-DCs infiltration between metastatic LNs and non-metastatic LNs was found. However, CD1a-DCs infiltration into LNs was significantly reduced after chemoradiotherapy when compared to the treatment-naive group [19].

In gastric cancer, no significant difference in CD1a-DCs infiltration was observed between metastatic LNs and non-metastatic LNs [36]. In patients with endometrial and cervical cancers, there were significantly higher numbers of CD1a-DCs in the sentinel LNs compared to the non-sentinel LNs, although no significant difference of CD1a-DCs infiltration was found between metastatic sentinel LNs and non-metastatic sentinel LNs [37].

CD209(DC-SIGN) is a well-known marker of DCs and is one member of the C-type lectin superfamily. It is not only a pattern recognition receptor but has also been implicated in immunoregulation of DCs [39]. Several studies have focused on the CD209-positive DCs in cancer tissue, such as cutaneous squamous cell carcinoma [40,41], oral squamous cell carcinoma [42], renal cell carcinoma [43,44], prostate intraductal carcinoma [45], breast cancer [46], hepatocellular carcinoma [47] and colorectal cancer [48]. We could find only one study that focused on the regional LNs of solid cancer. Yamada et al. [49] investigated the regional LNs of colorectal cancer and found that CD209 was strongly expressed in the sinus macrophages of LNs. Sinus macrophages are believed to possess antigen-presenting capabilities, are involved in anti-cancer immune responses, and the amount of sinus macrophages reduced by aging [50]. In this study, CD209-positive cells were found in the sinuses of LNs as previously reported [49], in both metastatic LNs and non-metastatic LNs. In contrast, CD1a-DCs were not detected in the sinuses of LNs and were only present in cancerous tissue of metastatic LNs.

The limitations of the present research include the small sample size (*n* = 70), the limited follow-up period (median follow-up time of 47.7 months), the retrospective nature of the study, and manual evaluation of DCs involves the problem of interobserver variation. Although we initially tried to assess conventional DCs using IHC of CD83 then IHC of CD86, assessment of CD83-positive cells and CD86-positive cells were difficult because their intensity revealed by IHC was faint and the dendritic shapes were unclear in the present study.

To the best of our knowledge, this is the first study that has focused on CD1a-DCs infiltration into regional LNs of GBC patients. The results indicate strong confounding between CD1a-DCs and LN metastasis and suggests the theory that CD1a-DCs are developed from monocytes at tumor sites. As the varied results of previous studies, which mainly focused on CD1a-DCs infiltration into regional LNs of other organs, the role and significance of CD1a-DCs infiltration into regional LNs may be different according to tumor histology or the primary tumor site. Studies which focused on CD1a-DCs infiltration into primary tumor and regional LNs were few in number and the evaluation methods of CD1a-DCs were not uniform. Therefore, the accumulation of further research findings is required to unequivocally clarify the role and significance of CD1a-DCs infiltration into regional LNs and primary tumor.

## 4. Materials and Methods

### 4.1. Patients

A total of 101 consecutive patients with GBC who underwent surgical resection of their primary lesion at Saga University Hospital between 2000 and 2020 were initially enrolled in the study. After the exclusion of patients with non-invasive or intramucosal cancer (pTis or pT1a), no LNs harvested, non-curative resection and synchronous advanced gastric cancer, a total of 70 patients with GBC were included (Figure 5). All patients provided informed consent for the use of resected tissue, and the study protocol was approved by the Ethics Committee of the Faculty of Medicine at Saga University (No. 2023–02-R-08). Clinical and histopathological staging were based on the TNM Classification of Malignant Tumors (8th edition) provided by the Union for International Cancer Control.

### 4.2. Immunohistochemistry

Immunohistochemistry (IHC) for CD1a and S100 was performed using a single representative block of formalin-fixed, paraffin-embedded tumor and LN specimens obtained from each patient. Sections (4 μm) were deparaffinized, and antigen retrieval was performed using Histofine Heat Processor Solution pH 9 (Nichirei Biosciences, Tokyo, Japan) and an automatically controlled thermostat (Histofine HEAT PROII; Nichirei Biosciences). The following primary antibodies were used: the mouse monoclonal anti-CD1a antibody (clone 010; IS06930–2; prediluted; Dako, Glostrup, Denmark); the rabbit polyclonal anti-S100 antibody (GA50461–2 J; prediluted; Dako); and the mouse monoclonal anti-CD209 antibody (clone 120507; R&D systems, Minneapolis, MN, USA). The Envision+ System (Dako) was used as the secondary antibody. Slides were stained with diaminobenzidine tetrahydrochloride, and nuclei were counterstained with hematoxylin. An Autostainer Plus (Dako) was used for the staining of specimens.

### 4.3. Assessment of CD1a-DCs and S100-DCs

IHC slides were scanned using NanoZoomer S360 (Hamamatsu Photonics, Shizuoka, Japan) to obtain digital images. CD1a-DCs and S100-positive DCs (S100-DCs) were manually evaluated by measuring hot spots at a ×100 magnification. Most of the CD1a-positive cells consistently showed a dendritic shape. As for S-100, monocytes and macrophages, which did not exhibit dendritic shapes, were also positive, so only those that had a dendritic shape were counted. In cases with LN metastasis, DCs in metastatic LNs were measured. Patient cohorts were divided into high- (>10/×100 magnification) and low-group (≤10/×100 magnification), both for regional LNs and primary tumor, according to a previous report [9].

### 4.4. Statistical Analysis

All statistical analyses were performed using JMP Pro ver. 13.1.0 software (SAS Institute, Cary, NC, USA). Student’s *t*-test, Pearson’s chi-squared test and a linear regression analysis were employed when appropriate to make comparisons between two groups. Overall survival (OS) was defined as the period from surgery to death or the last follow-up. Disease-specific survival (DSS) was defined as the period from surgery to cancer-related death or the last follow-up. Relapse free survival (RFS) was defined as the period from surgery to cancer relapse or the last follow-up. The maximum follow-up period in the study was 120 months, with a median follow-up time of 47.7 months. The survival curve was plotted using the Kaplan–Meier method and a log-rank test was conducted.

## Figures and Tables

**Figure 1 ijms-25-12763-f001:**
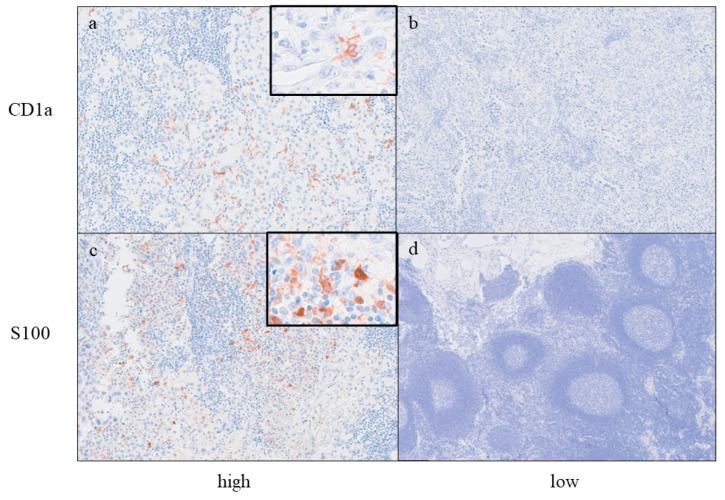
Representative images of immunohistochemistry of CD1a ((**a**) high group, (**b**) low group) and S100 ((**c**) high group, (**d**) low group). The original magnification of each photo was ×200, and the insets are enlarged images of high magnification of dendritic cells (original magnification ×400).

**Figure 2 ijms-25-12763-f002:**
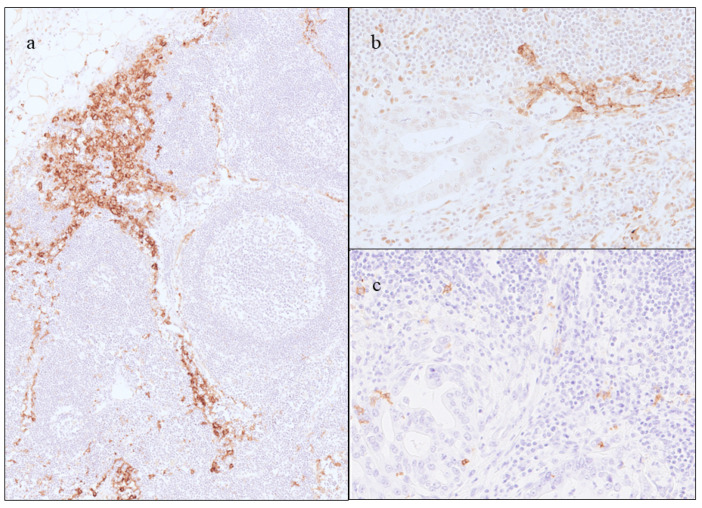
Representative images of immunohistochemistry of CD209 (DC-SIGN) ((**a**) original magnification ×100, (**b**) original magnification ×200). (**a**) Many CD209-positive cells are found at the sinuses of LNs. (**b**) In metastatic LNs, CD209-positive cells are found at stroma surrounding cancer cells, some of them having dendritic shapes. (**c**) Image of immunohistochemistry of CD1a, almost the same area of (**b**). Dendritic-shaped CD1a-positive cells are found at stroma surrounding cancer cells (original magnification ×200).

**Figure 3 ijms-25-12763-f003:**
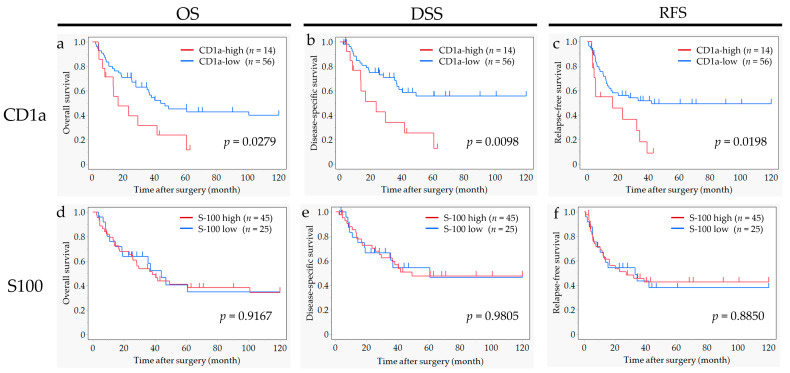
Overall survival (OS), disease-specific survival (DS) and relapse free survival (RFS) curves according to CD1a-positive dendritic cell infiltration into regional LNs (**a**–**c**), and S100-positive dendritic cell infiltration into regional LNs (**d**–**f**).

**Figure 4 ijms-25-12763-f004:**
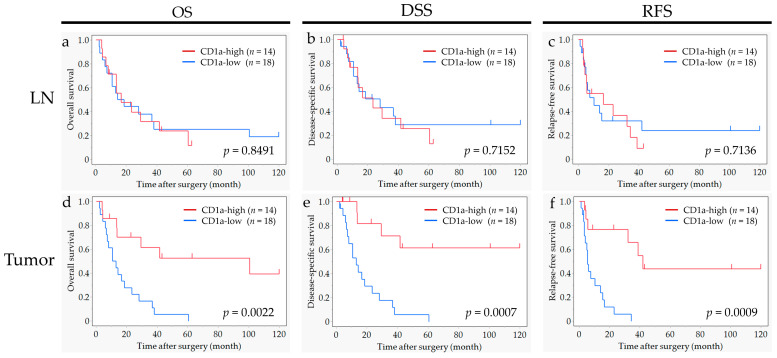
Subgroup analyses of cases with LN metastasis (*n* = 32). Survival curves for overall survival (OS), disease-free survival (DSS) and relapse-free survival (RFS) according to the status of CD1a-DC infiltration into regional lymph nodes (**a**–**c**) and the primary tumor (**d**–**f**) are shown. No significant impact of CD1a-DCs infiltration into regional lymph nodes was found on OS, DSS and RFS. In contrast, a significant impact of CD1a-DCs infiltration into the primary tumor was found for OS (*p* = 0.002), DSS (*p* = 0.001) and RFS (*p* = 0.001).

**Figure 5 ijms-25-12763-f005:**
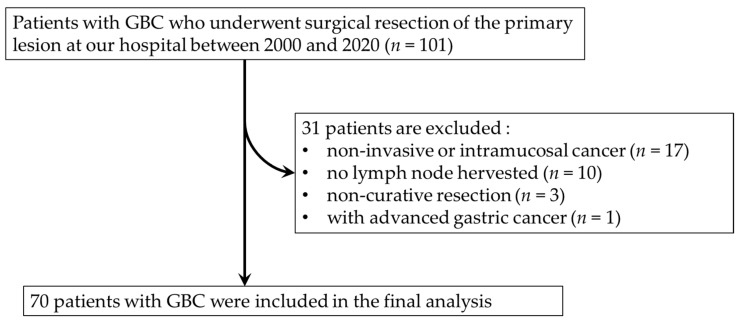
Flow chart showing case selection criteria.

**Table 1 ijms-25-12763-t001:** Clinicopathological features and the status of DCs in regional LNs of GBC (*n* = 70).

Age Years (Mean ± SD)		71.1 ± 8.8
Gender		
	Male	29 (41.4)
	Female	41 (58.6)
T (%)		
	1b	7 (10)
	2a	11 (15.7)
	2b	18 (25.7)
	3	28 (40)
	4	6 (8.6)
N (%)		
	N0	38 (54.3)
	N1	27 (38.6)
	N2	5 (7.1)
M (%)		
	M0	61 (87.1)
	M1	9 (12.9)
CD1a-DCs in LN		
	high	14 (20.0)
	low	56 (80.0)
S100-DCs in LN		
	high	45 (64.3)
	low	25 (35.7)
CD1a-DCs in tumor		
	high	35 (50.0)
	low	35 (50.0)
Adjuvant therapy		
	done	37 (52.9)
	not done	33 (47.1)

Abbreviations: DCs, dendritic cells; SD, standard deviation.

**Table 2 ijms-25-12763-t002:** Association between the clinicopathological features and status of CD1a-DCs infiltration into regional LNs (*n* = 70).

Characteristics	CD1a-DCs High (*n* = 14)	CD1a-DCs Low (*n* = 56)	*p*-Value
Age years (mean ± SD)	70.9 ± 8.66	71.9 ± 9.43	0.718
Male/Female (%)	6/8 (42.9/57.1)	23/33 (41.1/58.9)	1.000
T1b, T2/T3, T4 (%)	3/11 (21.4/78.6)	33/23 (58.9/41.1)	0.017
N0/N1, N2 (%)	0/14 (0.0/100.0)	38/18 (32.1/67.8)	<0.0001
M0/M1 (%)	11/3 (78.6/21.4)	6/50 (10.7/89.3)	0.370
CD1a-DCs in tumor (high/low)	8/6 (57.1/42.9)	27/29 (48.2/51.8)	0.766
Adjuvant therapy (done/not done)	8/6 (57.1/42.9)	29/27 (51.8/48.2)	0.772

Abbreviations: DCs, dendritic cells; SD, standard deviation.

**Table 3 ijms-25-12763-t003:** Association between the clinicopathological features and status of S100-DCs infiltration into regional LNs (*n* = 70).

Characteristics	S100-DCs High (*n* = 45)	S100-DCs Low (*n* = 25)	*p*-Value
Age years (mean ± SD)	70.9 ± 8.69	71.4 ± 9.05	0.821
Male/Female (%)	21/24 (46.7/53.3)	8/17 (32.0/68.0)	0.313
T1b, T2/T3, T4 (%)	24/21 (53.3/46.7)	12/13 (48.0/52.0)	0.804
N0/N1, N2 (%)	23/22 (51.1/48.9)	15/10 (60.0/40.0)	0.617
M0/M1 (%)	40/5 (88.9/11.1)	21/4 (84.0/16.0)	0.712
Adjuvant therapy (done/not done)	24/21 (53.3/46.7)	13/12 (52.0/48.0)	1.000

Abbreviations: DCs, dendritic cells; SD, standard deviation.

**Table 4 ijms-25-12763-t004:** Univariate analyses for DSS, OS and RFS for all patients (*n* = 70).

Label	OS		DSS		RFS	
	HR (95% CI)	*p*-Value	HR	*p*-Value	HR	*p*-Value
Age <71 years	1.204 (0.651–2.225)	0.554	1.680 (0.828–3.408)	0.151	1.498 (0.786–2.855)	0.220
Gender (male)	1.225 (0.661–2.274)	0.519	0.980 (0.475–2.021)	0.957	0.974 (0.505–1.879)	0.937
T3, T4	4.444 (2.242–8.809)	<0.000	7.431 (3.023–18.266)	<0.000	3.419 (1.709–6.839)	0.001
N1, N2	3.025 (1.605–5.670)	0.001	4.014 (1.882–8.564)	0.000	3.199 (1.636–6.254)	0.001
M1	3.501 (1.542–7.950)	0.003	4.986 (2.114–11.759)	0.000	6.006 (2.691–13.409)	<0.000
Low CD1a-DCs in tumor	1.636 (0.878–3.048)	0.121	2.159 (1.033–4.509)	0.041	1.530 (0.798–2.934)	0.201
High CD1a-DCs in LN	2.195 (1.890–4.423)	0.028	2.714 (1.272–5.790)	0.010	2.327 (1.144–4.733)	0.020
High S100-DCs in LN	1.035 (0.542–1.976)	0.917	0.991 (0.474–2.070)	0.981	0.952 (0.490–1.850)	0.885
Adjuvant therapy (done)	1.389 (0.739–2.612)	0.308	2.078 (0.953–4.530)	0.066	3.409 (1.597–7.281)	0.002

Abbreviations: HR, hazard ratio; CI, confidence interval; OS, overall survival; DSS, disease-specific survival; RFS, relapse free survival.

**Table 5 ijms-25-12763-t005:** Multivariate analyses of factors associated with OS, DSS and RFS (*n* = 70).

Label	OS		DSS		RFS	
	HR	*p*-Value	HR	*p*-Value	HR	*p*-Value
Age <71 years	0.884 (0.424–1.842)	0.742	1.050 (0.448–2.462)	0.911	1.143 (0.543–2.404)	0.725
T3, T4	3.741 (1.542–9.072)	0.004	5.560 (1.791–17.256)	0.003	1.585 (0.633–3.970)	0.326
N1, N2	1.393 (0.548–3.540)	0.486	1.183 (0.394–3.546)	0.765	1.363 (0.514–3.610)	0.534
M1	1.889 (0.738–4.833)	0.185	2.357 (0.878–6.328)	0.089	2.764 (1.240–7.460)	0.045
Low CD1a-DCs in tumor	1.521 (0.765–3.024)	0.232	1.769 (0.803–3.897)	0.157	1.208 (0.574–2.542)	0.619
High CD1a-DCs in LN	1.214 (0.542–2.722)	0.637	1.486 (0.610–3.620)	0.384	1.204 (0.520–2.784)	0.665
Adjuvant therapy (done)	0.734 (0.327–1.648)	0.454	0.826 (0.297–2.298)	0.715	2.006 (0.807–4.984)	0.134

Abbreviations: HR, hazard ratio; CI, confidence interval; OS, overall survival; DSS, disease-specific survival; RFS, relapse free survival.

**Table 6 ijms-25-12763-t006:** Univariate analyses for OS, DSS and RFS in the LN metastasis group (*n* = 32).

Label	OS		DSS		RFS	
	HR (95% CI)	*p*-Value	HR	*p*-Value	HR	*p*-Value
Age <71 years	1.666 (0.738–3.761)	0.219	1.920 (0.798–4.622)	0.145	2.068 (0.900–4.753)	0.087
Gender (male)	1.795 (0.245–1.269)	0.164	1.674 (0.671–4.176)	0.269	1.077 (0.441–2.627)	0.871
T3, T4	4.207 (0.976–18.136)	0.054	7.282 (0.967–54.849)	0.054	2.249 (0.663–7.628)	0.193
M1	2.498 (1.006–6.205)	0.049	3.566 (1.355–9.382)	0.010	3.159 (1.233–8.092)	0.017
Low CD1a-DCs in tumor	4.495 (1.716–11.774)	0.002	7.068 (2.282–21.898)	0.001	6.871 (2.207–21.390)	0.001
High CD1a-DCs in LN	1.081 (0.484–2.418)	0.849	1.173 (0.497–2.768)	0.715	1.166 (0.5103–2.664)	0.716
High S100-DCs in LN	1.532 (0.638–3.681)	0.340	1.186 (0.477–2.946)	0.714	1.083 (0.457–2.564)	0.856
Adjuvant therapy (done)	0.338 (0.149–0.769)	0.010	1.956 (0.745–5.136)	0.173	0.840 (0.328–2.153)	0.716

Abbreviations: HR, hazard ratio; CI, confidence interval; OS, overall survival; DSS, disease-specific survival; RFS, relapse-free survival.

## Data Availability

Data contained within the article can be obtained from the authors upon reasonable request.

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
