# Peer review of "Analysis of CD1a-Positive Monocyte-Derived Cells in the Regional Lymph Nodes of Patients with Gallbladder Cancer"

_ijms, 2024, doi:10.3390/ijms252312763_

Round 1

Reviewer 1 Report (Previous Reviewer 2)

Comments and Suggestions for Authors

I fully understand the effort the authors have put into revising the manuscript. However, I still firmly believe that the authors need to validate whether the CD1a+ cells are indeed dendritic cells (DCs). Based on the provided CD209 data, it seems to suggest so. However, the morphology of CD209+ cells partially resembling DCs does not confirm their identity as DCs.

I propose the following:

First, the authors need to demonstrate the co-expression of CD1a and S100 on DCs in tissue sections. They may refer to this article: https://www.sciencedirect.com/science/article/pii/S0006497120520752.

Additionally, the authors should provide a precise description of the percentage of DCs expressing CD1a and S100. Are all CD1a+ (or S100+) cells DCs, or do all DCs express CD1a or S100? If not, how do the authors distinguish CD1a+ DCs from non-DCs? These aspects should be thoroughly discussed.

Author Response

We would like to thank all the reviewers for their careful reading of our manuscript and for providing such useful feedback, which was a great help in improving the clarity of our manuscript. The changes based on the reviewer’s comments are highlighted (blue). 

Reviewer 1

Query/Comment: I fully understand the effort the authors have put into revising the manuscript. However, I still firmly believe that the authors need to validate whether the CD1a+ cells are indeed dendritic cells (DCs). Based on the provided CD209 data, it seems to suggest so. However, the morphology of CD209+ cells partially resembling DCs does not confirm their identity as DCs.

I propose the following:

First, the authors need to demonstrate the co-expression of CD1a and S100 on DCs in tissue sections. They may refer to this article: https://www.sciencedirect.com/science/article/pii/S0006497120520752.

Additionally, the authors should provide a precise description of the percentage of DCs expressing CD1a and S100. Are all CD1a+ (or S100+) cells DCs, or do all DCs express CD1a or S100? If not, how do the authors distinguish CD1a+ DCs from non-DCs? These aspects should be thoroughly discussed.

Response: Thank you for your valuable suggestion. The proposition “Whether CD1a positive cells are DCs or not" is a very important point of discussion. In recent years, it has become common to classify DCs into bone marrow-derived classical DCs (cDCs) and plasmacytoid DCs (pDCs), and also monocyte-derived DCs (moDCs). CD1a-positive cells are classified as moDC because they are differentiated from monocytes by cytokines such as TNFα, GM-CSF, IL-6, IL-4 and IL-1β. As the reviewer has pointed out, the debate still remains whether moDCs are essentially macrophages or DCs. However, the opinion that moDCs are not macrophages but DCs, and that moDCs and monocyte-derived macrophages (moMacs) are different, has become dominant in recent years. CD1a is recognized as a marker for moDCs (please refer to the following review, especially Table 2 and Box 1; Trends in Immunology, 2023, 44, 999-1013, https://doi.org/10.1016/j.it.2023.10.005). The original manuscript we submitted lacked an explanation of this background. We sincerely apologize for this oversight. In the revised manuscript, we have added this explanation to the Introduction with appropriate references. In addition, since the opinion that moDCs involving CD1a-positive cells are macrophages but not DCs still remains, we have amended the title, replacing the phrase “CD1a-positive dendritic cells” to “CD1a-positive monocyte-derived cells”. We have also added the sentence that "CD1a is known as a marker of monocyte-derived DCs" to the Abstract, removed S100 from the keywords and replaced it with the word monocyte.

Regarding confirmation of the co-expression of CD1a and S100, although S-100 is known as a classical DC marker, it is also expressed in monocytes. Although it may be possible to confirm co-expression by fluorescent double staining in one lymph node, it was unrealistic to do so for all 70 cases. Moreover, even if co-expression of CD1a and S100 was observed, this result does not necessarily resolve the essential question of whether CD1a-positive cells are DCs or macrophages. CD1a is considered as a more specific marker for moDC. In addition, the main purpose of the present study was “To examine the relationships between CD1a-positive cells and clinicopathological factors in regional LNs of gallbladder cancer". The purpose can be rephrased from the viewpoint of moDC thus: “To examine the situation in which monocytes differentiate into CD1a-positive moDC in regional lymph nodes of gallbladder cancer." Therefore, the findings regarding S100 and CD209 are not mainstream to our research. The confusion and misunderstanding caused was entirely due to our carelessness in describing "CD1a-positive DCs" without providing sufficient explanation. We deeply apologize for this inadvertent omission and would like to express our deep gratitude to the reviewer for his expert advice.

At the present time, we cannot fully respond to the reviewer's request. However, our result that "monocytes differentiated into CD1a-positive cells (moDCs) only in lymph nodes with metastasis" indicates that monocytes were differentiated into CD1a-DCs in lymph nodes by association with tumor cells. This finding is supported by evidence that moDCs have a low migratory capacity and do not express CCR7, which is the chemokine receptor required for migration to LNs. We believe that our findings will definitely contribute to future research on tumor immunity. We have added a description regarding these matters to the revised Discussion section.

Reviewer 2 Report (Previous Reviewer 1)

Comments and Suggestions for Authors

line 92: Representative histological figures of DCs at reginal LNs are shown in Figure 1.

please correct in all text typo like this

Author Response

We would like to thank all the reviewers for their careful reading of our manuscript and for providing such useful feedback, which was a great help in improving the clarity of our manuscript. The changes based on the reviewer’s comments are highlighted (blue). 

Reviewer 2

Query/Comment: line 92: Representative histological figures of DCs at reginal LNs are shown in Figure 1.

please correct in all text typo like this

Response: We have amended the text as following: “Representative histological figures of CD1a-DCs and S100-DCs at regional LNs are shown in Figure 1”.

We found other mistyping in Figure 4 (mistyping: CD1a/S100, corrected to LN/Tumor) and in the Material and Methods of “4.1. Patients” section (mistyping: Figure 4, corrected to Figure 5). We would like to thank the reviewer for their kind and helpful comments.

Reviewer 3 Report (New Reviewer)

Comments and Suggestions for Authors

This study addresses an important topic in the tumor microenvironment of gallbladder cancer, but several methodological issues must be addressed to improve the robustness and interpretation of the findings.

1. The use of CD1a and S100 as markers to define dendritic cells (DCs) is problematic due to their lack of specificity. CD1a is not exclusively expressed by DCs and may also be present in other cell types, such as certain B cells, while S100 is broadly expressed in neutrophils, macrophages, and other immune and non-immune cells. The authors should provide a justification for not including more classical and widely accepted markers such as CD45 and CD11c, which are standard in defining DCs. A combination of CD45, CD11c, and CD1a could offer a more robust approach to ensure the accurate identification of DCs.

2. The manuscript lacks sufficient detail about how DC morphology was assessed to distinguish these cells from other S100- or CD1a-positive cells. The authors should explicitly describe whether the identification of DCs was performed using algorithm-based automated methods, or manually. If manual identification was used, the authors should clarify the criteria employed, to promote reproducibility of the findings.

3. The survival analysis has several limitations that may compromise the reliability of the conclusions: Small sample size: The study includes only 70 patients, with 14 cases in the high CD1a group. This small sample size could lead to insufficient statistical power and increased susceptibility to outliers, which may bias the results. Confounding factors: The high expression of CD1a in lymph nodes is strongly correlated with nodal metastasis, suggesting a confounding relationship. This correlation weakens the conclusion that CD1a-DC infiltration independently impacts survival. Follow-up duration: The median follow-up of 47.7 months may not fully capture long-term survival outcomes, particularly for early-stage cases with better prognosis. Extended follow-up data would strengthen the conclusions. The authors should discuss these limitations more explicitly and, if feasible, consider reanalyzing the data with methods that mitigate these issues.

4. The field of gallbladder cancer research has advanced significantly in recent years, with several single-cell and bulk RNA-sequencing datasets now publicly available. The authors should validate their findings using these resources, (e.g., Gut PMID: 35868538 and Cell Discovery PMID: 35273221). Single-cell RNA-seq could provide insights into the transcriptional signatures of CD1a+ and S100+ cells in the tumor microenvironment.

Author Response

We would like to thank all the reviewers for their careful reading of our manuscript and for providing such useful feedback, which was a great help in improving the clarity of our manuscript. The changes based on the reviewer’s comments are highlighted (blue). 

Reviewer 3

Query/Comment 1: The use of CD1a and S100 as markers to define dendritic cells (DCs) is problematic due to their lack of specificity. CD1a is not exclusively expressed by DCs and may also be present in other cell types, such as certain B cells, while S100 is broadly expressed in neutrophils, macrophages, and other immune and non-immune cells. The authors should provide a justification for not including more classical and widely accepted markers such as CD45 and CD11c, which are standard in defining DCs. A combination of CD45, CD11c, and CD1a could offer a more robust approach to ensure the accurate identification of DCs.

Response: Monocyte-derived DCs (moDCs), including CD1a-positive cells, are thought to have intermediate characteristics between macrophages and DCs, and there is still debate as to whether moDCs are essentially macrophages or DCs. However, the opinion that moDCs are not macrophages but DCs, and that moDCs and monocyte-derived macrophages (moMacs) are different, has become dominant in recent years and CD1a is recognized as a marker for moDCs (please refer to the following review, especially Table 2 and Box 1; Trends in Immunology, 2023, 44, 999-1013, https://doi.org/10.1016/j.it.2023.10.005). In addition, as added to the Discussion section of the previously submitted manuscript, there are papers which consider that even CD209, which is well known as a dendritic cell marker, can be expressed on macrophages. Similarly, CD45 is positive in leukocytes in general, and CD11c is also known as a macrophage marker, so even if this staining was performed, it would not be possible to unequivocally conclude that CD1a-positive cells are DCs. The manuscript we previously submitted basically missed an explanation of this background material. In the revised manuscript, this explanation has now been added to the Introduction and Discussion along with appropriate references.

Query/Comment 2: The manuscript lacks sufficient detail about how DC morphology was assessed to distinguish these cells from other S100- or CD1a-positive cells. The authors should explicitly describe whether the identification of DCs was performed using algorithm-based automated methods, or manually. If manual identification was used, the authors should clarify the criteria employed, to promote reproducibility of the findings.

 Response: CD1a-positive cells were manually determined on digital slides scanned by NanoZoomer S360 (Hamamatsu Photonics, Shizuoka, Japan). CD1a-DCs and S100-DCs were manually evaluated by measuring hot spots at a ×100 magnification. Most of the CD1a-positive cells consistently showed a dendritic shape. As for S-100, monocytes and macrophages, which do not present with a dendritic shape, were also positive, so only those cells that had a dendritic shape were counted. These matters had been revised at “4.3. Assessment of CD1a-DCs and S100-DCs” in the Materials and Methods section of the revised manuscript. The dendritic shape is a reliable morphological feature to distinguish moDCs from macrophages. This matter is now considered in the Introduction of the revised manuscript (revised manuscript, lines 74-78). As we manually evaluated DCs, the problem of interobserver variation would remain. This point is additionally described as a limitation in the Discussion section of revised manuscript (lines 310-311). In addition, we have deleted the sentence “the assessment of mature DCs was only an indirect marker of S100” from the limitations because DC1a is sufficient for the evaluation of moDCs.

Query/Comment 3: The survival analysis has several limitations that may compromise the reliability of the conclusions: Small sample size: The study includes only 70 patients, with 14 cases in the high CD1a group. This small sample size could lead to insufficient statistical power and increased susceptibility to outliers, which may bias the results. Confounding factors: The high expression of CD1a in lymph nodes is strongly correlated with nodal metastasis, suggesting a confounding relationship. This correlation weakens the conclusion that CD1a-DC infiltration independently impacts survival. Follow-up duration: The median follow-up of 47.7 months may not fully capture long-term survival outcomes, particularly for early-stage cases with better prognosis. Extended follow-up data would strengthen the conclusions. The authors should discuss these limitations more explicitly and, if feasible, consider reanalyzing the data with methods that mitigate these issues.

 Response: As the reviewer wisely pointed out, the small sample size limits our statistical analysis. However, the number of cases and the follow up period are unimprovable issues, so there is no other way to conduct multi-institutional brand-new studies. These matters are described in the Discussion section as limitations of our study (revised manuscript, lines 308-309).

As the reviewer has pointed out, high CD1a-positive cell infiltration was only observed in cases of LN metastasis and therefore it is a confounding factor with survival. Our study does not conclude that high CD1a-positive cell infiltration in LN is an independent factor, but as shown in the multivariate analysis (Table 5), we conclude that “High CD1a-positive cell infiltration in LN is not an independent factor for survival". We believe that this "strong confounding between lymph node metastasis and CD1a positive cells" is rather a strong point of our study, indicating that monocytes were differentiated into CD1a-positive cells in LNs by the tumor cells.

Query/Comment 4: The field of gallbladder cancer research has advanced significantly in recent years, with several single-cell and bulk RNA-sequencing datasets now publicly available. The authors should validate their findings using these resources, (e.g., Gut PMID: 35868538 and Cell Discovery PMID: 35273221). Single-cell RNA-seq could provide insights into the transcriptional signatures of CD1a+ and S100+ cells in the tumor microenvironment.

Response: Thank you for your valuable suggestion. There are a paucity studies that have focused on the association of human cancer tissue and CD1a-positive cells and therefore it is unclear how cancer cells trigger the differentiation of monocytes into CD1a. As monocytes differentiate into CD1a-positive cells triggered by TNFα during inflammation, it is a possible hypothesis that the TNFα produced by tumor cells promotes differentiation of them into CD1a-positive cells. However, this is too complex a theme to do additional experiments in the present study and would be difficult to achieve with the research resources we currently have available. There are many unknowns about the role of CD1a-positive cells in tumor immunity, and if this important issue is clarified by future studies, it would be a striking and important discovery for this field of knowledge.

Round 2

Reviewer 1 Report (Previous Reviewer 2)

Comments and Suggestions for Authors

No more comments

Reviewer 3 Report (New Reviewer)

Comments and Suggestions for Authors

I have no additional questions towards this manuscript. The authors have answered my questions. 

This manuscript is a resubmission of an earlier submission. The following is a list of the peer review reports and author responses from that submission.

Round 1

Reviewer 1 Report

Comments and Suggestions for Authors

the paper presents the infiltration score of CD1a DCs in gallbladder cancers regional LNs (n=70) and analyzes data using Kaplan Meyer and univariate and multivariate statistics. It states unfavoureble outcome (only by univariate analysis) for cases with high infiltration (equal or more than 10 cells in 100x hotspots); these cases are all from patients bearing metastasis. The results were not in accordance with the literature and to what the authors expected to and it is probably due to the amount of cases (only 14 with high CD1a DCs infiltration) or the method chosen to quantify cells

major points: 

- some mistakes in the text (line 127)

- low quality images (the number of 100x hotspots counted was not stated). Please show not overexposed photos. I find difficult to count at 100x of magnification, the count could be underextimated. Compare hotspots and case with lesser content of CD1a and indicate metastasis if present.

- in table 4, 5 and 6 appear the group of low CD1a DCs in tumor. But this is not the case because the count was performed only in tumor draining lymph node, isn't it? Also in line 193 and 195 is stated that CD1a DCs are in tumor...

- it's very strange that S100 stains lesser than CD1a

- line 207 rewrite, CD1a DCs fundamental? is a too strong sentence

- Rephrase sentence 208-211. 

Comments on the Quality of English Language

English is ok but some sentences are to rephrase.

Author Response

Responses to reviewers

We would like to thank all the reviewers for their careful reading of our manuscript and for providing such useful feedback, which was a great help in improving the clarity of our manuscript. We have responded to all of the queries and made all the requested changes to the revised manuscript. All changes based on the reviewer’s comments are highlighted. The English usage and grammar of the revised manuscript have been kindly checked by a senior editor at Oxford University who is familiar with its contents.

Our point-by-point responses to the reviewer’s comments are given below.

Reviewer 1

Query/Comment 1: Regarding some mistakes in the text (line 127).

Response: We have corrected the mistakes in that sentence (in the low high CD1a-DC → in the high CD1a-DC).

Query/Comment 2: Regarding quality of Figure 1 and overexposed photos.

Response: We have replaced the photos in Figure 1 to normal exposure images. In the revised Figure 1, DCs are more easily recognized.

Query/Comment 3: Regarding counting CD1a-DCs in tumor in Table 4, 5, 6.

Response: We have counted CD1a-DCs not only in lymph nodes but also in primary tumor (S-100 DCs at the tumor were not counted because S100 DCs did not produce a significant clinical impact in our previous study). This finding is documented in our previous manuscript on lines 106-108 and lines 302-303; however, it might be unkind for readers and misleading due to insufficient explanation. In the revised manuscript, we have therefore added documentation regarding the CD1a-DCs in tumor in lines 83, 91-93, 316-317 and in Table 1.

Query/Comment 4: Regarding S100-DCs seemed less than CD1a-DCs in Figure 1.

Response: In the revised version of Figure 1, S100-DCs are more easily recognized and S100 DCs occurred more frequently than CD1a-DCs.

Query/Comment 5: Regarding the rephrasing sentences at line 208-211 (including the line 207 “fundamental”).

Response: We have rephrased these sentences that sound redundant as follows:

“We initially considered that LN infiltration by CD1a-DCs would be associated with favorable clinical outcomes and that if CD1a-DCs migrated into regional LNs from the primary site, there would be CD1a-DCs in not only metastatic LNs but also LNs without cancer metastasis”.

Once again, we would like to thank all the reviewers for their time and effort put in in to greatly improving the clarity of our manuscript.

Reviewer 2 Report

Comments and Suggestions for Authors

In this manuscript, the authors examined the infiltration of CD1a-positive dendritic cells in the regional lymph nodes of patients with gallbladder cancer, finding that their presence was associated with poor clinical outcomes but was not an independent prognostic factor. In contrast, CD1a-positive dendritic cell infiltration in the primary tumor significantly impacted surgical outcomes, suggesting their role may vary depending on tumor location and type.

As mentioned by the authors in the discussion section, this study has limitations in terms of sample size, technical constraints, and the depth of mechanistic investigation. It also lacks sufficient innovation and clinical relevance. Moreover, my primary concern is that using IHC staining for CD1a or S100 alone is not sufficient to conclusively identify dendritic cells (DCs). At the very least, immunofluorescence staining with multiple markers should be used to definitively confirm the presence of DCs before any further analysis. Based on this, I find the persuasiveness of this article to be insufficient.

Author Response

Responses to reviewers

We would like to thank all the reviewers for their careful reading of our manuscript and for providing such useful feedback, which was a great help in improving the clarity of our manuscript. We have responded to all of the queries and made all the requested changes to the revised manuscript. All changes based on the reviewer’s comments are highlighted. The English usage and grammar of the revised manuscript have been kindly checked by a senior editor at Oxford University who is familiar with its contents.

Our point-by-point responses to the reviewer’s comments are given below.

Reviewer 2

Query/Comment 1: Regarding the limitations of IHC for DC markers.

Response: As reviewer 2 requested we have now described limitations of the study in the revised Discussion, the small number of DC markers examined being a weakness. Especially, analyses of multiple mature DC markers (such as CD83 and CD86) would be beneficial. Initially, we tried to evaluate mature DC by CD83 IHC but the positive stain of CD83 IHC was faint, and the dendritic shapes of CD83-positive cells were unclear. Thus, we failed to evaluate CD83 IHC. Although we considered that evaluation of mature DCs is difficult except for flow cytometry, the idea of immunofluorescence provided by reviewer 2 is a very nice suggestion. As we do not have enough time and research resources at the present time, additional immunofluorescence analyses could not be performed; however, we will undoubtedly incorporate in our next study immunofluorescence of multiple DC markers.

Although the current study may not be conclusive, there are few studies on cancer and DCs using pathological specimens, and no studies have been reported on regional LNs of gallbladder cancer. Even in its present form, we believe that out study will be useful and of great interest to researchers investigating the association between cancer and DCs.

Once again, we would like to thank all the reviewers for their time and effort put in in to greatly improving the clarity of our manuscript.

Round 2

Reviewer 1 Report

Comments and Suggestions for Authors

The work has been improved.

Reviewer 2 Report

Comments and Suggestions for Authors

The authors' responses have not resolved any of my concerns or doubts. Discussion and explanation alone are insufficient to support their arguments, and these issues affect the reliability of the study. Therefore, I still recommend rejecting the manuscript.